# A Polypyrrole-Modified Pd-Ag Bimetallic Electrode for the Electrocatalytic Reduction of 4-Chlorophenol

**Xuefeng Wei** [1,*] , **Laiyuan Zeng** [1] , **Weiwei Lu** [1,*] , **Juan Miao** [1,2] , **Ruichang Zhang** [1,2] , **Ming Zhou** [1,2] **and Jun Zhang** [1]

1   College of Chemical Engineering & Pharmaceutics, Henan University of Science and Technology, Luoyang 471023, China; zenglaiyuan@163.com (L.Z.); miaojuan@haust.edu.cn (J.M.); ruichangzhang@126.com (R.Z.); axore@163.com (M.Z.); j-zhang@126.com (J.Z.)
2   Luoyang Key Laboratory of Soil Pollution Remediation Project, Henan University of Science and Technology, Luoyang 471023, China
*   Correspondence: xfwei@haust.edu.cn (X.W.); luweiwei@haust.edu.cn (W.L.); Tel.: +86-379-6423-1914 (X.W.); Fax: +86-379-6562-7502 (X.W.)

**Abstract:** A polypyrrole-modified bimetallic electrode composed of Pd-Ag on a Ti substrate (Pd-Ag/PPY/Ti) was successfully prepared via a chemical deposition method, and was applied to the electrocatalytic hydrodechlorination of 4-chlorophenol (4-CP) in aqueous solution. The electrode was characterized by cyclic voltammetry (CV), scanning electron microscopy (SEM), X-ray diffraction (XRD), and X-ray photoelectron spectroscopy (XPS). Various influences on the dechlorination efficiency of 4-chlorophenol, including applied current, initial pH value, and temperature, were studied. The dechlorination efficiency of 4-CP reached 94% within 120 min under the optimum conditions, i.e., a dechlorination current of 6 mA, an initial pH of 2.30, and a temperature of 303 K. The apparent activation energy of the dechlorination of 4-CP by the Pd-Ag/PPY/Ti electrode was calculated to be 49.6 kJ/mol. The equivalent conversion rate constant $k_{Pd}$ was 0.63 L.$g_{Pd}^{-1}$·$min^{-1}$, which was higher than the findings presented in comparable literature. Thus, a highly effective bimetallic electrode with promising application prospects and low Pd loading was fabricated.

**Keywords:** electrocatalytic hydrodechlorination; Pd-Ag; bimetallic electrode; electroless plating; 4-chlorophenol

## 1. Introduction

Chlorophenols (CPs) are significant chemical intermediates for the manufacture of fungicides, antiseptics, dyes, pesticides, and personal-care-products in various industries [1–4]. The high toxicity and poor biodegradability of CPs has classified them in the blacklist of priority pollutants announced by European regulatory authorities and the US Environmental Protection Agency [5–7]. The residues of these chemicals are exposed to the air, water, and soil from industrial wastewater emission due to their global manufacture and purchase, and often have toxicological effects on human beings, such as skin and mucosa irritability, soft tissue sarcomas, and exposure via the contamination of groundwater and soil [2,8,9]. The toxic property of these chemicals is in part due to the chloride content in the molecules. In general, the increased number of chlorine atoms in their structures leads to the aggravation of toxicity related to CPs [10]. Hence, developing the path towards safe disposal and decreasing the threshold level of CPs in the final effluent is a requirement for ensuring public health security.

In recent years, various approaches have been proposed for removing CPs from effluents, including biodegradation [8], electron beam treatment [11], chemical oxidation [12], reductive dechlorination [13], thermal treatment [14], adsorption [15], and photodegradation [8].

Among the proposed technologies, electrocatalytic hydrodehalogenation (ECH) has been regarded as a feasible method for the removal of halogen atoms in chlorine-containing aromatic hydrocarbons [16–18] due to its ease of operation, simple reactor, mild reaction conditions, low secondary pollution risk, and high efficiency [19–21]. The efficiency of ECH depends on the competition between hydrogenation and hydrogen evolution reactions (HER). The competition of these two processes is influenced by several factors: (i) the substrate (e.g., the concentration and molecular structure of the substrate [22]); (ii) the reaction conditions (e.g., the applied current density/potential, the supporting electrolyte, the solution pH, and the temperature [23,24]); and (iii) the cathode material (including Ag [22], Pt, Au [25], Cu [26], and Pd [27]). Consequently, employing high-performance electrodes and optimizing the operation conditions have been utilized to achieve high ECH efficiency [17]

Among the factors that affect ECH, metal materials are inevitable components in cathode catalysts, which play a key role in the ECH reaction. For a high-performance ECH catalyst, metal catalysts should have a strong bond with hydrogen to ensure the proton–electron transfer process, but reduce the overpotential to allow the dissociative electron transfer of C–Cl bonds and avoid the HER process [22,28]. Pd has been considered as an ideal catalyst for ECH due to its lower binding energy with H and higher capacity for hydrogen adsorption and storage over other metals [17,21]. During ECH reaction, large quantities of $H_{ads}$ produced by the electroreduction of water or hydronium ions are adsorbed on the Pd catalyst, which increases the ECH efficiency of organic chlorides in water [17]. However, the major drawbacks of these noble metals are their high price and ease of agglomeration.

A series of efforts has been made to solve these problems, including the design of alloys or the development of auxiliary materials to reduce the agglomeration of Pd nanoparticles (NPs) [29–31]. Secondary non-noble metals, such as Ni [32–34], Cu, and Ag [35], have been introduced in palladium catalysts, as this is expected to reduce the Pd dosage and significantly boost the electro-catalytic performance of Pd NPs compared to monometallic Pd. This is due to the favorable adjustment of the electronic structure and a downward shift of the Pd d-band center based on the bimetallic synergistic effect [36]. Therefore, secondary metals act as promoters along with the Pd NPs during the ECH process. The kind of optimized cathode material has been recently widely reported. For instance, a Pd-Ni/Ti electrode exhibited the highest dechlorination efficiency (almost 100% removal) for the ECH of 2-chlorophenol as compared to Pd/Ti and Ni/Ti cathodes [32]. Moreover, He et al. observed that Pd/Ag/Ni and Pd/Cu/Ni electrodes had higher catalytic performance and current efficiencies compared with a Pd/Ni electrode for 2-chlorobiphenyl dechlorination [35].

Catalyst support materials are required to possess a considerable surface area, chemical stability, and superior electrical conductivity, which ensure that chlorinated organic molecules can easily access the catalytic sites and thus dramatically promote the efficiency of ECH. Carbon-based materials, such as carbon felt [37], carbon black, carbon nanotubes (CNTs) [1,6,38], and graphene [17,39,40], or conductive polymers, such as polymeric pyrrole (PPY) [27,33], poly (3,4-ethylenedioxythiophene) (PEDOT) [41], and polyaniline [42], have been generally used as supports.

Among these support materials, PPY has attracted enormous consideration in recent years due to its high electrical conductivity, good electrochemical properties, nontoxic properties, and ease of synthesis [43]. In addition, it can well disperse Pd particles and facilitate the availability of active sites on the catalytic interface, and thus enhance catalytic activity [27,44,45]. In this work, a PPY-modified Pd-Ag (Pd-Ag/PPY/Ti) composite cathode was fabricated for ECH. The Pd-Ag bimetallic catalysts were loaded via the electroless deposition method. The prepared electrode was characterized via CV, SEM, XRD, and XPS. In the ECH experiment to evaluate the electrocatalytic activity of the composite bimetallic electrodes, 4-chlorophenol (4-CP) was selected as the targeted contaminant.

## 2. Results and Discussion

### 2.1. Optimization of Preparation Conditions of Electrodes

#### 2.1.1. Effect of the Composition of the Plating Solutions

According to the mechanism of ECH [46], the current value of hydrogen adsorption/desorption in the CV curve is closely related to the transition of hydrogen adsorption/desorption on the electrode [45]. A series of electrodes was prepared in different plating solutions and characterized by CV. The compositions of the plating solutions and the hydrogen adsorption current values of the Pd-Ag/PPY/Ti electrodes, which were prepared in plating solutions with different concentrations of $PdCl_2$ and $AgNO_3$, are listed in Table 1.

**Table 1.** Variations of hydrogen adsorption current values of the Pd-Ag/PPY/Ti electrodes prepared with composition of the plating solution *.

| Composition (mmol/L) | $PdCl_2$ | 2 | 2 | 0 | 1 | 1 | 4 | 2 | 0.5 |
|---|---|---|---|---|---|---|---|---|---|
| | $AgNO_3$ | 1 | 0 | 1 | 1 | 1.5 | 1 | 2 | 0.5 |
| **Hydrogen Adsorption Current (mA)** | | 80 | 21 | 30 | 60 | 56 | 75 | 40 | 62 |

\* Reaction conditions: $C_{EDTA}$ = 40 g/L, $C_{N2H4}$ = 5 mol/L, $T$ = 333 K, $t$ = 2 h.

As can be seen in Table 1, the electrode prepared by mixing a 2 mmol/L plating solution of $PdCl_2$ and a 1 mmol/L solution of $AgNO_3$ had a higher hydrogen adsorption current value than the electrodes prepared in a pure 2 mmol/L solution of $PdCl_2$ or a pure 1 mmol/L plating solution of $AgNO_3$. This indicates that the catalytic activity of the Pd-Ag bimetal was better than that of the single Ag electrode and Pd electrode. In addition, the hydrogen adsorption current value of the Pd-Ag/PPY/Ti electrode was the largest of the other plating solution components. Therefore, it is appropriate to control the concentrations of $PdCl_2$ and $AgNO_3$ in the plating solutions to be 2 mmol/L and 1 mmol/L, respectively.

#### 2.1.2. Effect of Reaction Time

To investigate the effects of different reaction times on the Pd-Ag/PPY/Ti electrodes, the time was set as 1, 1.5, 2, 2.5, and 3 h, respectively. The hydrogen adsorption current values of the electrodes prepared at different reaction times were characterized by CV, and are presented in Figure 1.

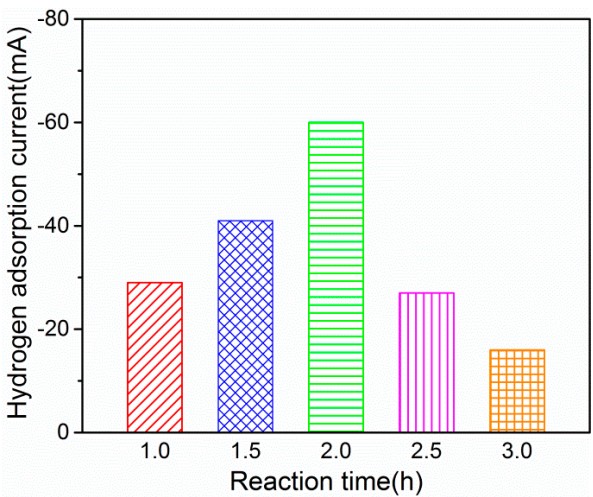

**Figure 1.** Effect of different reaction times on the Pd-Ag/PPY/Ti electrode.

From Figure 1, it is evident that the hydrogen adsorption current value increased gradually with the prolongation of deposition time (0–2 h), indicating that more Pd and Ag were deposited on the PPY/Ti electrode. However, the hydrogen adsorption current value decreased significantly when the deposition time exceeded 2 h. This may be due to the agglomeration deposition of Pd and Ag on the electrodes, which resulted in the catalysts falling off from the electrode and the reduction of the specific surface area of the catalysts; this coincides with the experimental phenomenon to some extent.

### 2.2. Characterization of Electrodes

#### 2.2.1. Cyclic Voltammetry Analysis

Figure 2 presents the CV curves of the PPY/Ti and Pd-Ag/PPY/Ti electrodes in 0.1 mol/L sulfuric acid solution obtained with a scan rate of 50 mV/s.

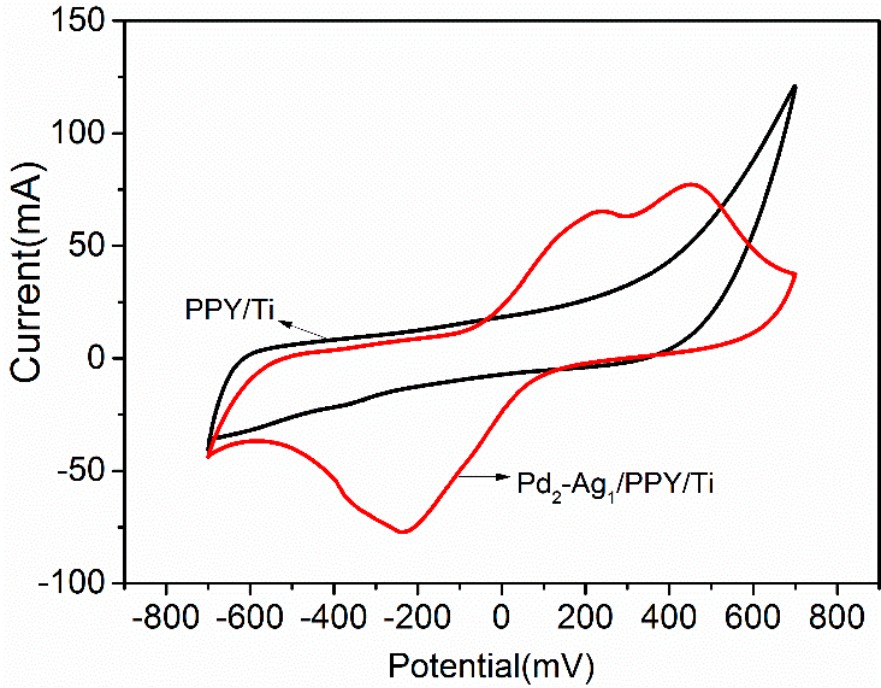

**Figure 2.** CV curves of the PPY/Ti and Pd-Ag/PPY/Ti electrodes.

Moreover, the PPY-modified film was conducive to the subsequent adhesion of Pd and Ag particles on the electrode. The Pd-Ag/PPY/Ti electrode had obvious hydrogen and oxygen adsorption/desorption peaks, which indicates that the Pd-Ag/PPY/Ti electrode had a strong hydrogen adsorption capacity and good catalytic reduction dechlorination potential.

#### 2.2.2. Morphology Analysis

The morphologies of the electrodes treated by different procedures were examined via SEM. As illustrated in Figure 3a, the surface of the PPY/Ti electrode was spongy after the organic polymer polypyrrole was loaded on the titanium mesh, which increased the surface area of the titanium mesh and was beneficial to the loading and dispersion of the Pd and Ag particles. As can be seen in Figure 3b, the PPY/Ti electrode was fully etched, and its surface was ravined after fluorination. This property improved the hydrophilicity of the electrode, ensured the bonding strength between the coating and PPY/Ti electrode, and made it easier for metal ions to adsorb onto the PPY/Ti electrode in subsequent procedures. In Figure 3c, it is evident that $Pd^{2+}$ was reduced during the sensitization and activation steps, and was deposited uniformly on the surface of PPY/Ti in the form of $Pd^0$. Thus, the dense activation center was formed, which was conducive to the subsequent growth of the metal catalyst and

the formation of the Pd-Ag coating. As shown in Figure 3d, Pd and Ag particles were interwoven to create a uniform dendritic coverage on the surface of the supporting matrix, forming an abundant groove concave–convex structure that greatly increased the specific surface area of the catalyst on the matrix surface and promoted the catalytic reaction.

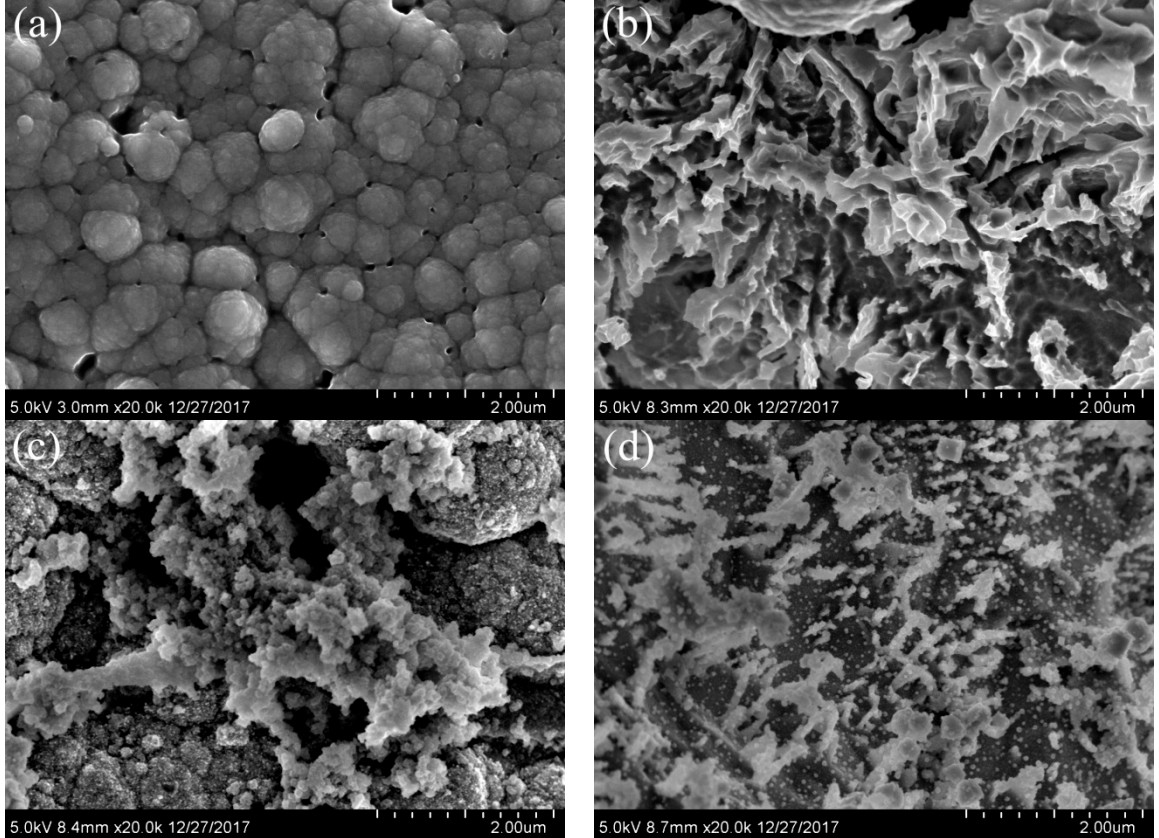

**Figure 3.**    SEM images of the modified electrodes:    (**a**) PPY/Ti electrode;    (**b**) PPY/Ti electrode after fluorination; (**c**) PPY/Ti electrode after fluorination, sensitization, and activation; (**d**) Pd-Ag/PPY/Ti electrode.

### 2.2.3. Analysis of Pd Content in the Electrode

The Pd content of the prepared electrode was determined via ICP-AES, and it was calculated that the Pd loading on the electrode was 1.13 mg. In addition, the energy-dispersive spectrometer (EDS) results (shown in Figure S1) confirmed the existence of Pd and Ag with a Pd:Ag ratio of 73.88:26.12, which indicates that both palladium and silver were successfully deposited onto the Ti mesh substrate.

### 2.2.4. XRD Analysis

The crystal structure of the Pd-Ag/PPY/Ti electrode was characterized by XRD, and is depicted in Figure 4. The patterns show the face-centered cubic (FCC) structure of Pd and Ag. The characteristic peak observed at $2\theta = 38.8°$, which was between the diffraction peaks of silver and palladium, corresponded to the Pd-Ag (111) alloy [47]. The symbols $IM_A$ and $IM_B$ are representative of impurities, perhaps Pb and Ni, respectively. In addition, the XRD patterns show amorphous phase characteristics because of the conducting polymer PPY interlayer on the electrode.

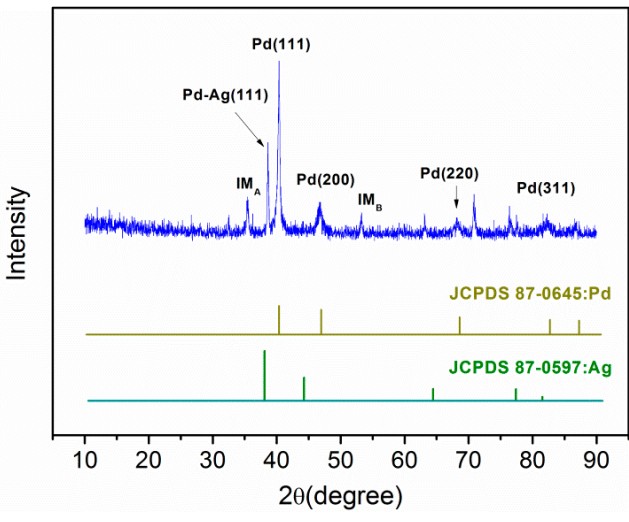

**Figure 4.** XRD pattern of the Pd-Ag/PPY/Ti electrode.

### 2.2.5. XPS Analysis

The surface species of the Pd-Ag/PPY/Ti electrode were further evaluated via XPS analysis (Figure 5a–d). Figure 5a confirms the presence of O, N, Ag, Pd, and C elements in the electrode. Additionally, the existence of the S element was attributed to the addition of dodecylbenzene sulfonate (SDBS) during the preparation of the electrodes.

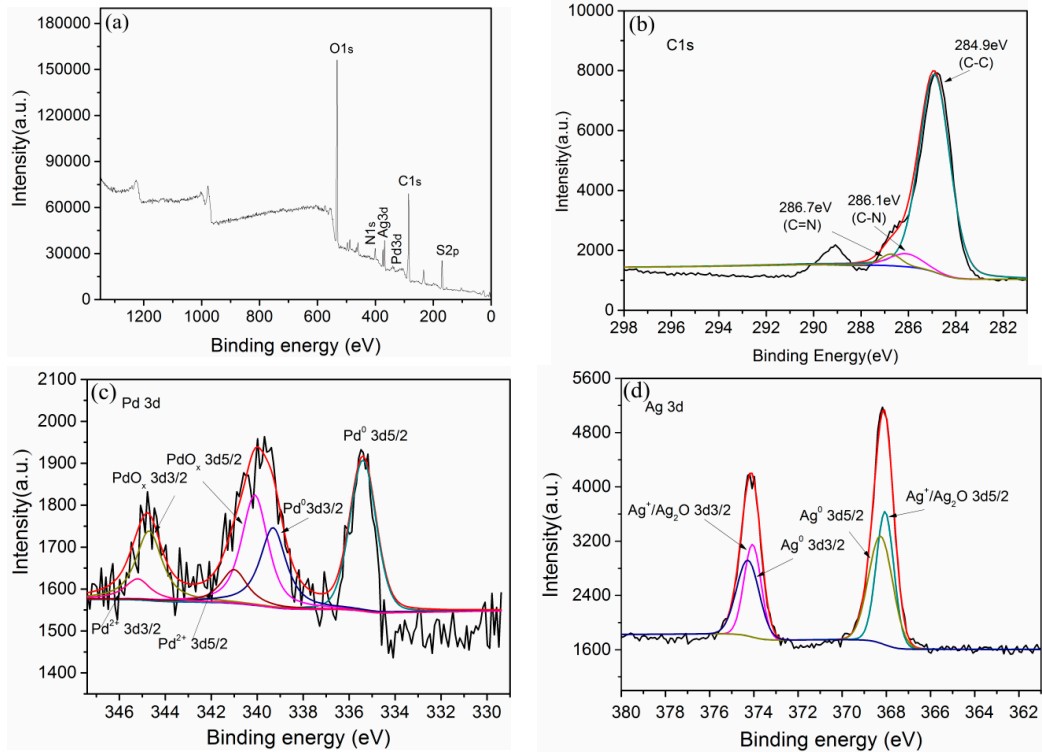

**Figure 5.** XPS of (**a**) full spectrum, (**b**) C 1s, (**c**) Pd 3d, and (**d**) Ag 3d of the Pd-Ag/PPY/Ti electrode.

The C 1 s spectrum shown in Figure 5b was fitted with three components. The three peaks corresponded to the structures of C–C, C–N, and C=N in the PPY films at binding energies of 284.9 eV, 286.1 eV, and 286.7 eV, respectively [48]. Moreover, the Pd 3d spectrum (Figure 5c) exhibited three pairs of spin-orbit components: (i) the binding energies of the spin-orbit split doublet (Pd 3d 5/2 and Pd 3d3/2) of 335.4 eV and 339.3 eV can be attributed to metallic palladium $Pd^0$, which confirms the

Pd deposition onto the surface of the electrode [49]; (ii) the binding energies of the 3d 5/2 and 3d 3/2 characteristic peaks were respectively 340.1 eV and 344.7 eV, and can be assigned to the PdOx species [50,51]; and (iii) the higher binding energy of the minor spin-orbit component pair indicates the presence of $Pd^{2+}$, which may be the residuum of $PdCl_2$ [52].

It can also be observed in Figure 5d that the Ag 3d peaks at 368.3 eV and 374.3 eV corresponded to $Ag^0$. The other pair of peaks at 368.1 eV and 374.0 eV corresponded to $Ag^+$ [50], which may be due to the partial oxidation of $Ag^0$ during the transfer to the spectrometer [30]. The binding energies of Pd 3d and Ag 3d both exhibited a slight shift according to their standard binding energies, because the electronegativity of Pd is 2.2, which is higher than the 1.8 electronegativity of silver, which led to the strong ability of Pd to attract electrons. Thus, the electrons around Ag migrated to Pd, and electron exchange occurred when the Pd-Ag alloys were formed [53].

### 2.3. ECH of 4-Chlorophenol

To evaluate the electrocatalytic activity of the prepared Pd-Ag/PPY/Ti electrode, experiments of the electrocatalytic reduction dechlorination of 4-CP were conducted, and the effects of constant current, initial pH, and temperature were investigated.

#### 2.3.1. ECH of 4-Chlorophenol

The current has a significant influence on the removal efficiency of 4-CP in aqueous solution. The electrolytic experiment of 4-CP was conducted under different constant currents (2 mA, 4 mA, 6 mA, and 8 mA) at room temperature. The initial concentration of 4-CP was 100mg/L, and the concentration of the supporting electrolyte $Na_2SO_4$ was 0.05 mol/L. The pH of the catholyte was 6.48 and was not adjusted.

Figure 6 presents the 4-CP removal efficiency versus time at different constant currents. Approximately 50% of 4-CP was removed by the Pd-Ag/PPY/Ti electrode at a reaction time of 120 min under a constant current of 2 mA. A significant period of time is required to produce $H_{ads}$ and migrate to the surface of the catalytic electrode when the current is low, which led to a lower dechlorination efficiency. The removal rate of 4-CP increased from 50% to 62% when the current increased from 2 mA to 6 mA, which indicates that dechlorination efficiency can be improved by increasing the current. However, the removal rate of 4-CP decreased to 47% when the current increased to 8 mA; this is because the rate of the side reaction of hydrogen evolution exceeded that of electrocatalytic hydrodechlorination, resulting in a decrease in the amount of $H_{ads}$ used for hydrodechlorination [54]. Therefore, the current value should be appropriate in order to obtain a higher dechlorination efficiency.

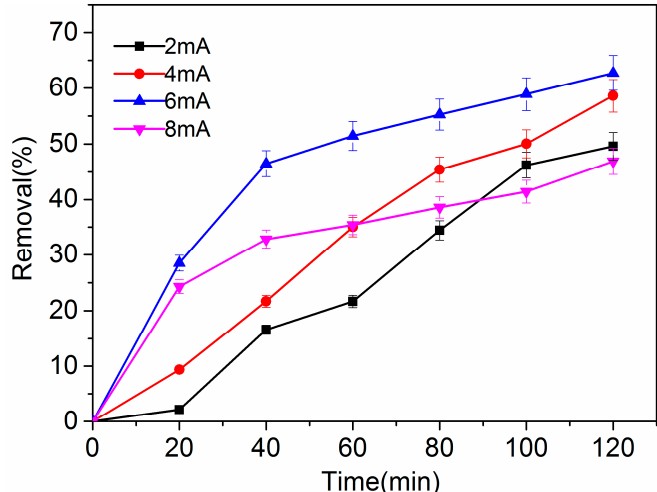

**Figure 6.** Effect of dechlorination current on the removal efficiency of 4-CP, pH = 6.48.

### 2.3.2. Effect of Initial pH Value

The effect of different initial pH values (2.05, 2.30, 2.53, and 6.48) on the removal efficiency of 4-CP was investigated under a current of 6 mA and time of 90 min at room temperature. According to the existing literature [29,55], 4-CP exists in the form of negative ions and is repulsed by electrostatic force when the solution is alkaline, which will lead to a low removal efficiency. On the contrary, numerous $H_{ads}$ will be generated and occupy active sites on the electrode surface in an acidic solution. Thus, only acidic conditions were investigated in the present study.

Figure 7 presents the effect of initial pH on the removal efficiency of 4-CP, and it is evident that the removal efficiency varied greatly at different initial pH values. The removal efficiency was only 62% at the initial pH value of 6.48; however, at pH values of 2.05, 2.30, and 2.40, the removal efficiencies were 91%, 88%, and 82%, respectively. This result indicates that, in a certain range, the removal efficiency was much higher when the pH of the catholyte was much lower.

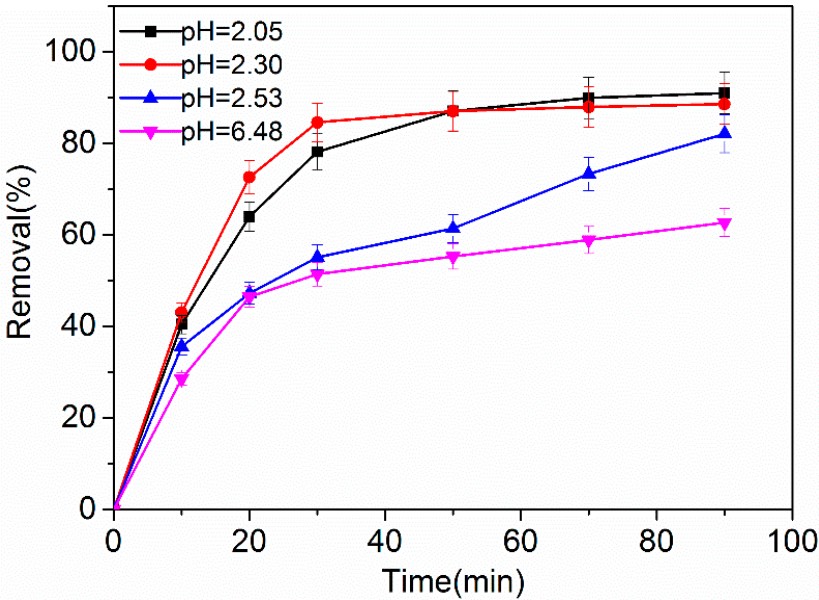

**Figure 7.** Effect of initial pH on the removal efficiency of 4-CP, I = 6 mA.

According to the mechanism of ECH, hydrogen ions were consumed during dechlorination, leading to the increase of the catholyte pH. The initial and terminal pH values are listed in Table S1. The $H_{ads}$ was insufficient and the dechlorination reaction proceeded slowly when the pH of the catholyte was not adjusted to acidity. However, a much lower pH value can lead to acid residue, which will increase the difficulty of subsequent processing.

### 2.3.3. Effect of Temperature

The kinetics of the hydrodechlorination reaction of 4-CP were studied. Figure 8 presents the fitting lines of $ln(C_t/C_0)$ against time under various reaction temperatures. The good linear correlation indicates that the hydrodechlorination of 4-CP followed pseudo-first-order kinetics. The apparent reaction rate constant $k_{obs}$ was equal to the slope of the fitting line, and was 0.00445, 0.00583, 0.01802, and 0.02383 min$^{-1}$ at 278 K, 288 K, 298 K, and 303 K, respectively. The slope of the fitted curve was low at 278 K and 288 K, indicating that the removal efficiency of 4-CP increased slowly. When the temperature increased to 298 K and 303 K, the removal rate of 4-CP increased rapidly, and the *k* value was 4 to 5 times that at 278 K and 288 K. In addition, the removal rate of 4-CP attained 94% at 303 K after 2 h of reaction, as the increase of temperature accelerated the mass transfer process of 4-CP in the solution. Thus, the reaction between $H_{ads}$ and 4-CP was accelerated.

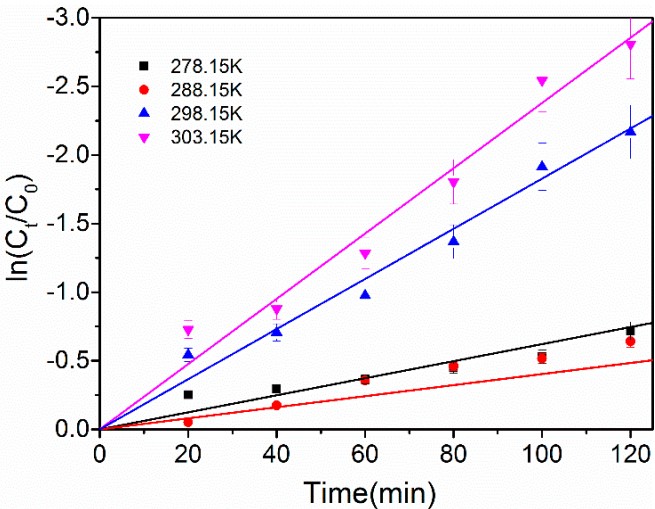

**Figure 8.** Linear fitting curves of 4-CP dechlorination at different temperatures, I = 6 mA, pH = 2.30.

The reaction activation energy of 4-CP on the surface of the Pd-Ag/PPY/Ti electrode can be calculated according to the Arrhenius equation, as shown in Equation (1):

$$k = Aexp(-Ea/RT),\tag{1}$$

where *k* is the apparent reaction rate constant of 4-CP on the electrodes (min$^{-1}$), *A* refers to the pre-factor, which is obtained by drawing, *Ea* is the activation energy of the reaction (kJ/mol), *R* is the molar gas constant 8.314 J/(mol·K), and *T* is the temperature of the reaction (K). The Arrhenius plots of *lnk* versus 1/*T* for the 4-CP dechlorination are shown in Figure S2, and the activation energy was calculated to be 49.6 kJ/mol.

According to Pd loading on the electrode via ICP-AES measurement, the equivalent conversion rate constant $k_{Pd}$ was calculated to be 0.63 L.$g_{Pd}^{-1}$·min$^{-1}$ at 303 K. To evaluate the catalytic activity and exclude the effect of different technologies and model pollutants, the kinetic parameters normalized by the Pd loading per liter reaction liquor (L.$g_{Pd}^{-1}$·min$^{-1}$) are listed in Table 2. Compared with the reported technologies in Table 2, the conversion rate constant in this work was higher, which indicates that the prepared Pd-Ag/PPY/Ti bimetallic composite cathode had better catalytic performance.

**Table 2.** Comparison of equivalent conversion rate constants under different technologies.

| Technology | Electrode or Catalyst | Model Pollutant | Pd Loading (mg.cm$^{-2}$) | $k_{obs}$ (min$^{-1}$) | $k_{pd}$ (L.$g_{Pd}^{-1}$·min$^{-1}$) | Reference |
|---|---|---|---|---|---|---|
| Electrocatalytic hydrodechlorination | Pd/Ni foam | 2,4-D | 1.78 | $0.5 \times 10^{-2}$ | 0.019 | [56] |
| | Pd/MnO$_2$/Ni foam | 2,4-DCBA | 0.44 | $1.5 \times 10^{-2}$ | 0.283 | [57] |
| | TiC-Pd/Ni foam | 2,4-DCBA | 0.44 | $3.9 \times 10^{-2}$ | 0.591 | [58] |
| | Pd-Ti/TiO$_2$NTs | TCE | 3.2 | $1.9 \times 10^{-2}$ | 0.37 | [59] |
| | Pd-Ag/PPY/Ti | 4-CP | 0.14 | $2.4 \times 10^{-2}$ | 0.63 | This work |
| Catalytic hydrodechlorination | Fe$_3$O$_4$@PPY@Pd nanoparticles | 4-CP | - | $2.3 \times 10^{-2}$ | 0.43 | [60] |
| | Pd/Fe$_3$O$_4$@SiO$_2$@m-SiO$_2$ | 4-CP | - | $3.5 \times 10^{-2}$ | 0.13 | [61] |
| | Fe/Pd | DCP | - | $4.0 \times 10^{-2}$ | 0.10 | [62] |
| Photocatalytic reduction | Pd/g-C$_3$N$_4$ | 2-CDD | - | $2.8 \times 10^{-3}$ | 0.06 | [63] |

## 3. Materials and Methods

### 3.1. Reagents and Materials

Analytical-grade palladium chloride (99.97%), silver nitrate (99.8%), sulfuric acid (98%), hydrochloric acid (36–38%), tin chloride (98%), ammonium hydroxide (25–28%), sodium fluoride (98%), hydrazine hydrate (40%), ethylene diamine tetraacetic acid (EDTA) (99.8%), sodium dodecylbenzene

sulfonate (SDBS) (99.8%), 4-chlorophenol (4-CP) (99%), and sodium sulfate (99.8%) were purchased from Sino pharm Chemical Reagent Co., Ltd. (Shanghai, China). Pyrrole (Py) was refined via vacuum distillation. Methanol and acetic acid were of chromatographic grade. A Pt sheet, Ti mesh, and $Hg/Hg_2SO_4$-saturated $K_2SO_4$ reference electrode were supplied by CH Instruments, Inc., Shanghai, China. Nafion-324 (DuPont, Midland, MI, USA) was used as the cation-exchange membrane. A quantity of 0.5 mol/L $H_2SO_4$ aqueous solution was used for CV tests and to adjust the pH value in the electrolysis reaction. All the solutions were prepared in deionized (DI) water obtained using a Millipore-Q system.

*3.2. Methods*

3.2.1. Preparation of Electrodes

Ti mesh substrate was pretreated prior to electrodeposition. It was cut to the desired size (2 cm × 3 cm) and then immersed in 0.3 mol/L $Na_2CO_3$ solution at approximately 333 K for 30 min to remove grease. The acid pickling process was accomplished in 0.1 mol/L oxalic acid at 333 K for 30 min. Finally, it was washed with DI water and stored in ethanol for further use. The treated Ti mesh was electrodeposited in a mixture solution of 0.2 mol/L $H_2SO_4$, 3 mmol/L SDBS, and 100 mmol/L distilled pyrrole under a 10-mA current for 5 min. Subsequently, the PPY/Ti electrode was kept in a vacuum drying chamber at 393 K for 30 min to obtain a homogeneous and adherent PPY film on the Ti mesh.

Prior to the chemical deposition of palladium and silver particles, a series of surface modification procedures was carried out on the PPY/Ti electrodes, including fluorination, sensibilization, activation, and neutralization reactions. The treatment steps and conditions are listed in Table 3.

**Table 3.** Steps of surface pretreatment for PPY/Ti electrodes*.

| Step | Reagent | Time (min) |
|---|---|---|
| Fluorination | 0.724 g/L $Na_2F$ and 1.5 mL/L HF | 15 |
| Sensibilization | 10 g/L $SnCl_2$ and 12 mL/L 36% HCl | 1 |
| Activation | 0. 25 g/L $PdCl_2$ and 2.5 mL/L 36% HCl | 1 |
| Neutralization | 1% ammonium hydroxide | 3 |

* The sensibilization, activation and neutralization processes were repeated 10 times.

The surface-modified PPY/Ti electrodes were immersed in a freshly prepared chemical plating solution that included ammonia (240 mL/L), EDTA (40 g/L), $N_2H_4$ (5 mol/L), and different concentrations of $PdCl_2$ and $AgNO_3$ at 333 K for 2 h. The obtained Pd-Ag/PPY/Ti electrode was then thermally treated at 393 K for 30 min. A fresh electrode was prepared and used in every dechlorination experiment.

3.2.2. Dechlorination Experiment

All the ECH experiments of 4-CP were performed in a two-compartment H-cell divided by a Nafion-324 membrane. A direct-current supply source was employed to offer a constant current. The Pd-Ag/PPY/Ti electrode and platinum sheet served as the cathode and anode, respectively. The anolyte was 30 mL of 50 mmol/L $Na_2SO_4$. The catholyte was a 30-mL mixed solution of 50 mmol/L $Na_2SO_4$ and 100 mg/L 4-CP with different pH values, and the catholyte underwent constant vigorous stirring during electrolysis to facilitate mass transfer. At 20-min intervals, 1 mL of the sample that passed through the 0.22-μm filter membrane was withdrawn from the catholyte for further analysis. The experimental equipment for dechlorination is shown in Figure S3.

3.2.3. Analytical Methods

The surface morphology and elemental/crystallite phases of the Pd-Ag/PPY/Ti electrode were determined via a scanning electron microscope (SEM, SU8020, Hitachi Limited, Tokyo, Japan) equipped with an energy-dispersive spectrometer (EDS, EX250, HORIBA, Kyoto, Japan), and X-ray diffraction (XRD, D8 ADVANCE, Bruker, Bremen, Germany) using Cu Kα radiation at a scan rate of 6°/min in the

2θ range of 10°–90°. X-ray photoelectron spectroscopy (XPS) was performed with a Thermo ESCALAB 250Xi system (Thermo Fisher Scientific, Waltham, MA, USA) using monochromatized aluminum as the X-ray source. The amounts of palladium and silver in the Pd-Ag/PPY/Ti electrode were confirmed by an inductively coupled plasma-atomic emission spectrometer (ICP-AES) (ICPE-9820, SHIMADZU, Kyoto, Japan). A potentiostat (CHI660, CH Instruments, Inc., Shanghai, China) and a DC-regulated power supply (DJS-292, Shanghai Rex Instrument Co., Ltd., Shanghai, China) were used to perform cycle voltammetry (CV) and electrolysis experiments, respectively. The CV experiments were carried out by a three-electrode system composed of an $Hg/Hg_2SO_4$ reference electrode, a Pt counter electrode, and a Pd-Ag/PPY/Ti working electrode.

The concentrations of 4-CP and its degradation products (phenol) were measured by a high-performance liquid chromatographer (HPLC, P230II, Elite, Shanghai, China) equipped with a C18 column (150 mm × 4.6 mm × 5 μm) and a UV detector. Isocratic elution was conducted with a mobile phase consisting of methanol/$H_2O$ (55:45, *v/v*) at a flow rate of 0.8 mL/min. The injection volume was 20 μL and the column temperature was 308 K. The spectrophotometric detection was performed at 280 nm.

The removal rate of 4-CP was calculated by Equation (2):

$$\eta = \frac{C_0 - C_t}{C_0} \times 100\%, \tag{2}$$

where $\eta$ is the removal rate of the target pollutant, $C_0$ is the initial concentration, and $C_t$ is the reaction concentration at time t (min).

The equivalent conversion rate constant $k_{Pd}$ was used to reflect the catalytic performance of the composite electrodes, and was calculated by Equation (3):

$$k_{pd} = \frac{k_{obs} \times V}{m}, \tag{3}$$

where $k_{Pd}$ is the equivalent conversion rate constant of the target pollutants on the electrodes ($L \cdot g_{Pd}^{-1} \cdot min^{-1}$), $k_{obs}$ is the apparent reaction rate constant of the target pollutants ($min^{-1}$), $V$ is the volume of the electrolyte (L), and $m$ is the equality mass of the Pd catalyst on the electrode (g).

## 4. Conclusions

A novel Pd-Ag/PPY/Ti electrode with high catalytic activity was successfully prepared via a chemical deposition process. The preparation conditions of the Pd-Ag/PPY/Ti electrodes, including the composition of the deposition solution, the deposition time, and the fluorination process, were optimized. The optimized Pd-Ag/PPY/Ti electrode was employed for the electrochemical reduction of 4-CP with a the current of 6 mA and initial pH of 2.30 at 303 K, and it was found that the removal rate attained 94% after 2 h of reaction time. The activation energy was calculated to be 49.66 kJ/mol, and the equivalent conversion rate constant $k_{Pd}$ was calculated to be 0.63 $L \cdot g_{Pd}^{-1} \cdot min^{-1}$. The prepared Pd-Ag/PPY/Ti electrode with lower Pd content had a good catalytic reduction efficiency of 4-CP, exhibiting a good application prospect for the electrocatalytic degradation of halogenated organic pollutants.

**Supplementary Materials:** The following are available online at http://www.mdpi.com/2073-4344/9/11/931/s1, Figure S1: EDS spectra of Pd-Ag/PPy/Ti electrode. Figure S2: The Arrhenius plot of *lnk* and 1/*T* of 4-CP dechlorination. Figure S3: Schematic digram of electrocatalytic hydrodechlorination equipment. Table S1: The pH value of catholyte before and after dechlorination process.

**Author Contributions:** Conceptualization, X.W.; Methodology, X.W.; Validation, W.L.; Formal Analysis, J.M.; Investigation, L.Z.; Writing-Original Draft Preparation, L.Z.; Writing-Review & Editing, M.Z.; Supervision, W.L. and X.W.; Project Administration, R.Z.; Funding Acquisition, J.Z.

**Funding:** This work was supported by National Natural Science Foundation of China (21403058, 21576073, and 41601520), the Key Scientific Research Projects of Higher Education Institutions in Henan Province (20A610001),

Natural Science Foundation of Henan province (182300410110) and Research Foundation for the Young Core Instructor Program of Henan Province, China (2016GGJS-058).

**Acknowledgments:** I acknowledge the support of Henan University of Science and Technology.

**Conflicts of Interest:** The authors declare no conflict of interest.

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
