# Peer review of "A Polypyrrole-Modified Pd-Ag Bimetallic Electrode for the Electrocatalytic Reduction of 4-Chlorophenol"

_catalysts, doi:10.3390/catal9110931_

Round 1

Reviewer 1 Report

This manuscript deals with the electrocatalytic reduction of 4-chlorophenol with a polypyrrole-modified Pd/Ag electrode. The work is sound, clearly written and describes the preparation of the electrode, its characterization (CV, SEM, XRD, XPS) and its efficiency to reduce 4-chlorophenol which presents public health problems.

I recommend the acceptation of this article because it is correctly conducted. It can be qualified as a routine contribution since the concept reported is not new and can be considered as an incremental work. However the catalytic results obtained are better than those previously reported for this reduction in the literature and deserves publication.

Before final acceptation of the manuscript, I would like to recommend the following minor revisions:

Page 3, line 117 to line 120: the description given seems to be subjective and not supported by strong scientific data or results and could be suppress. This part does not meet the scientific standard of a journal like Catalysts. The description of Fig 1 which is given in page 4 line 121-127 is more relevant. Page 4, line 134: I do not think that it is possible to qualify the CV as a parallelogram. What does this description mean? The CV curve of the PPY/Ti electrode exhibits the front in reduction and oxidation, not more. Page 5, line 163: The Pd loading should be given with a concentration. What does 1.13 mg of Pd mean? Page 9, line 266: this information is given for a second time. In fact, ICP-AES gives a concentration of Pd, which can be in ppm for example or in % weight. Page 8, line 240 : typo : dechlorination Some relevant references could, to my opinion, be cited in this manuscript:

Chemical Engineering Journal (Amsterdam, Netherlands) (2019), 358, 903-911

Toxicological & Environmental Chemistry (2016), 98(3-4), 327-344

Journal of Hazardous Materials (2018), 356, 17-25

Electrochimica Acta (2015), 178, 92-100

Electrochemistry Communications (2004), 6(3), 268-272

Journal of hazardous materials (2015), 290, 1-8

Author Response

Response to Reviewer 1

Page 3, line 117 to line 120: the description given seems to be subjective and not supported by strong scientific data or results and could be suppress. This part does not meet the scientific standard of a journal like Catalysts. The description of Fig 1 which is given in page 4 line 121-127 is more relevant. Page 4, line 134: I do not think that it is possible to qualify the CV as a parallelogram. What does this description mean? The CV curve of the PPY/Ti electrode exhibits the front in reduction and oxidation, not more.

Response: We fully agree with the reviewer. (i) We have deleted the description about the phenomenon of experimental process (line 117 to line 120 ), which does not meet the scientific standard; (ii) We have corrected the description of CV curve of the PPY/Ti electrode (line 134). “The CV curve of the PPY/Ti electrode is similar to a parallelogram, and exhibits obvious capacitance characteristics.” has been cortected as “The CV curve of the PPY/Ti electrode shows that the conductive polymer PPY can be reducted in lower potenial and oxidated in higer potenial, and exhibits capacitance characteristics”

Page 5, line 163: The Pd loading should be given with a concentration. What does 1.13 mg of Pd mean? Page 9, line 266: this information is given for a second time. In fact, ICP-AES gives a concentration of Pd, which can be in ppm for example or in % weight.

Response: Befor the ICP-AES measurement for Pd and Ag content in the electrode, we dissolved the metal catalysts using diluted nitrohydrochloric acid. The concertration of Pd ions in solution was detected in ppm and the taotal Pd laoding on the electrode was calculated. (i) Line 163 in the original manuscript, “it was found that the Pd loading was 1.13 mg” has been rewriten as , “it was calculated that the Pd loading on the electrode was 1.13 mg”;(ii) Page 9, line 266, in the original manuscript, the repated information was deleted. According to ICP-AES measurment, the Pd loading was 1.13 mg” has been written as “According to Pd loading on the electrode via ICP-AES measurment”.

Page 8, line 240 : typo : dechlorination

Response: We thank reviewer for pointing out our typo, it has been corrected.

Some relevant references could, to my opinion, be cited in this manuscript:

Chemical Engineering Journal (Amsterdam, Netherlands) (2019), 358, 903-911

Toxicological & Environmental Chemistry (2016), 98(3-4), 327-344

Journal of Hazardous Materials (2018), 356, 7-25

Electrochimica Acta (2015),178, 92-100

Electrochemistry Communications (2004), 6(3), 268-272

Journal of hazardous materials (2015), 290, 1-8

Response: We have supplemented the relevant references.

Reviewer 2 Report

Referee´s report:

The manuscript *A polypyrrole-modified Pd-Ag bimetallic electrode for the electrocatalytic reduction of 4-chlorophenol*, by  X. Wei, L. Zeng, W. Lu, J. Miao, R. Zhang, M. Zhou and J. Zhang  describes the design and preparation of a new electrode and its application in the reduction of chlorophenol. The interest in this work stems from the fact that chlorophenols, chemical intermediates used in several industries, e.g. the manufacture of fungicides, antiseptics, dyes, pesticides, and personal-care-products, are toxic and have poor biodegradability. Residues of these substances can contaminate humans by contamination of ground water and soil, and they have been classified as primary pollutants by several regulatory authorities. Hence the development of methods to destroy them and convert them into other less toxic or harmless substances is of great interest. In this manuscript the authors explain the problem involved in their introduction, the advantages of using electrocatalytic dehydrohalogenation (ECH) to modify this pollutant into a less harmful substance and give a brief literature survey.

They describe the preparation of a new electrode that they developed, a Pd(II)-Ag(I)/PPY/Ti electrode characterize it morphologically, by cyclic voltammetry, scanning electron microscopy, X-ray diffraction, and X-ray photoelectron spectroscopy (XPS), and analyze of its palladium content by ICP-AES. PPY is a conductive polymeric material consisting of a polymeric form of pyrrole. It has been used previously as a support in the design of other electrodes. Subsequently they optimize its operating conditions for the electrocatalytic dehydrohalogenation of chlorophenol.

The optimum ratio of Pd to Ag in the plating solutions was determined as well as the optimum reaction time for metal deposition on the electrode. The Pd to Ag ratio present at the end was determined by ICP-AES.

Operating parameters such as current, the initial pH value and temperature for the dechlorination were studied.

It was found that the electrode prepared as described in this manuscript could achieve 94% dechlorination efficiency within 120 min with a current of 6 mA, an initial pH of 2.30 at a temperature of 303 K. The conversion rate constant  (KPd) was determined to be 0.63 L.gPd-1.min.-1, which is a higher value than those described for other electrodes in the literature, according to the authors. They provide a table comparing the characteristsics of several electrodes or catalysts and the respective literature references.

They also provide a method to clean the electrode after operation.

This manuscript is written in a clear manner, the electrode developed was well characterized and the optimization studies appear clear.

The work described should be of interest to others working in this field and I recommend it for publication in Catalysts, after minor revision. I point out a few errors which should be corrected, but the authors should also look again throughout the manuscript for any small additional errors which may occur, particularly language errors.

Corrections which will improve readability:

Line 37: *chloride atoms * => *Chlorine atoms*

Line 46: *mild reaction * => *mild reaction conditions*

Line 54: *operation conditions * => *operating conditions *

Line 61: *massive Hads produced * => *large quantities of Hads produced*

Line 83: *electrical property * => *electrical properties *

Line 93: *2.1.1. Effect of plating solutions components *=> *2.1.1. Effect of the composition of the platting solutions*

After determining the optimum composition for the platting solution, the authors refer to the electrode as Pd2-Ag1/PPY/Ti electrode. This should be changed to Pd-Ag/PPY/Ti electrode wherever it occurs thoughout the text, and it should be simply mentioned that the electrode was prepared with a 2 mmol/L solution of PdCl2  and a 1 mmol/L solution of AgNO3. For example, the sentence

*As can be seen in Table 1, the Pd2-Ag1/PPY/Ti electrode prepared by the mixed plating 102 solutions of PdCl2 and AgNO3 had a higher hydrogen adsorption current value than the Pd2/PPY/Ti 103 and Ag1/PPY/Ti electrodes prepared in pure PdCl2 and AgNO3 plating solutions * should read something like:

*As can be seen in Table 1, the electrode prepared by mixing a 2 mmol/L plating solution of PdCl2 and a 1 mmol/L solution of AgNO3 had a higher hydrogen adsorption current value than the electrodes prepared in a pure 2 mmol/L solution of PdCl2 or a pure 1 mmol/L plating solution of AgNO3.*

In all the references, after one name, there should be a semi-colon followed by a space before the next name is written, i.e.

Reference 1:

Xiong, J.;Ma, Y.;Yang, W.;Zhong, L. S.Rapid, highly efficient and stable … should be Xiong, J.; Ma, Y.; Yang, W.; Zhong, L. S

 As well as for all the others

There should also be a space between the title and the initials of the authors.

This applies to every reference.

In addition, some references have every word capitalized, and this should not be so, e.g. ref. 52.

Author Response

Response to Reviewer 2

Corrections which will improve readability:

Line 37: “chloride atoms ” => “Chlorine atoms”

Line 46: “mild reaction ” => “mild reaction conditions”

Line 54: “operation conditions” => “operating conditions ”

Line 61: “massive Hads produced ” => “large quantities of Hads produced”

Line 83: “electrical property ” => “electrical properties ”

Line 93: “2.1.1. Effect of plating solutions components ”=> “2.1.1. Effect of the composition of the platting solutions”

Response: We thanks reviewer for their impatient and helpful suggestions.We have corrected them one by one.

After determining the optimum composition for the platting solution, the authors refer to the electrode as Pd2-Ag1/PPY/Ti electrode. This should be changed to Pd-Ag/PPY/Ti electrode wherever it occurs thoughout the text, and it should be simply mentioned that the electrode was prepared with a 2 mmol/L solution of PdCl2 and a 1 mmol/L solution of AgNO3. For example, the sentence

“As can be seen in Table 1, the Pd2-Ag1/PPY/Ti electrode prepared by the mixed plating solutions of PdCl2 and AgNO3 had a higher hydrogen adsorption current value than the Pd2/PPY/Ti and Ag1/PPY/Ti electrodes prepared in pure PdCl2 and AgNO3 plating solutions” should read something like:

“As can be seen in Table 1, the electrode prepared by mixing a 2 mmol/L plating solution of PdCl2 and a 1 mmol/L solution of AgNO3 had a higher hydrogen adsorption current value than the electrodes prepared in a pure 2 mmol/L solution of PdCl2 or a pure 1 mmol/L plating solution of AgNO3.”

Response: Reviewer’s suggestions are helpful. We have corrected it. All the “Pd2-Ag1/PPY/Ti electrode”have been changed to “Pd-Ag/PPY/Ti electrode” in the revised manuscript.

In all the references, after one name, there should be a semi-colon followed by a space before the next name is written, i.e.

Reference 1:

Xiong, J.;Ma, Y.;Yang, W.;Zhong, L. S.Rapid, highly efficient and stable … should be Xiong, J.; Ma, Y.; Yang, W.; Zhong, L. S

As well as for all the others

There should also be a space between the title and the initials of the authors.

This applies to every reference.

Response: We thanks reviewer for the corrections. The spaces between the semi-colon and next author were added. The spaces between author and title were added, too.

In addition, some references have every word capitalized, and this should not be so, e.g. ref. 52.

Response: We have cerrected it in Ref. 52 and checked carefully all the references.The same mistakes in other references have been modified.

Reviewer 3 Report

Dear Authors, it was good to read a well-described and proper designed article. Nevertheless, there are some crucial details ought to be reconsidered.

1. The formation of Pd-Ag layer should be better described. The way a deposition was performed often has a rather strong impact on the final material. Was it made step by step, i.e. Ag after Pd?

Powder XRD part: the text and the data have to be analyzed again. It's wrong in the present form.

2. The reference [45] doesn't contain any Pd-Ag pattern as was mentioned in the text. Please, check that and all other references to be sure is it fine, or not.

3. A typical Ag pattern (fcc structure) you can find e.g. via Google.

https://images.app.goo.gl/Gc8aDbJ3vi1g5R41A

The pattern you're presenting in Fig. 4 is not the right one, as far as there is no strongest line at the low-angle region. Silver fcc powder XRD pattern looks similar to Pd, but the lines are shifted due to another unit cell parameter.

4. The interpretation of XRD, in general, should be improved: please, do mark all the lines correctly. If there are unknown lines at the pattern, it should be marked by any symbol as an "unknown phase" or "impurity". If possible, do find a professional from the XRD field to help make this part well described.

I hope the remarks will help you to improve the article. Best wishes.

Author Response

Response to Reviewer 3

The formation of Pd-Ag layer should be better described. The way a deposition was performed often has a rather strong impact on the final material. Was it made step by step, i.e. Ag after Pd?

Powder XRD part: the text and the data have to be analyzed again. It's wrong in the present form.

Response: (i) The Pd-Ag layer was prepared via the electroless deposition method in “one-pot ”. The electroless plating solution consists of PdCl2, AgNO3, EDTA ,etc. So, it can be thought that the Pd and Ag ions be reduced by N2H4 almost at the same time; (ii) We thanks reviewer for pointing out our mistakes ,and we rewritten the XRD part.

The reference [45] doesn't contain any Pd-Ag pattern as was mentioned in the text. Please, check that and all other references to be sure is it fine, or not.

Response:Thanks reviewers’ correction, the reference [45] is wrong, it has been corrected in the revised manuscript. We have checked other references.

A typical Ag pattern (fcc structure) you can find e.g. via Google.

https://images.app.goo.gl/Gc8aDbJ3vi1g5R41A

The pattern you're presenting in Fig. 4 is not the right one, as far as there is no strongest line at the low-angle region. Silver fcc powder XRD pattern looks similar to Pd, but the lines are shifted due to another unit cell parameter.

Response: Thanks reviewers’ correction, we have redrawn the Figure of XRD.

The interpretation of XRD, in general, should be improved: please, do mark all the lines correctly. If there are unknown lines at the pattern, it should be marked by any symbol as an "unknown phase" or "impurity". If possible, do find a professional from the XRD field to help make this part well described.

Response: Thanks reviewers’ correction, we have redrawn the Figure of XRD and supplemented the impurity mark. Because the conducting polymer PPY interlayer was loaded on the electrode before the Pd and Ag deposition, the XRD patterns show amorphous phase characteristics, burrs existing.